# Adherence and the Diabetic Foot: High Tech Meets High Touch?

**DOI:** 10.3390/s23156898

**Published:** 2023-08-03

**Authors:** Hadia Srass, J. Karim Ead, David G. Armstrong

**Affiliations:** 1Southwestern Academic Limb Salvage Alliance, Department of Surgery, Keck School of Medicine of University of Southern California, 1450 San Pablo St #6200, Los Angeles, CA 90033, USA; 2College of Engineering, University of California, Riverside, 900 University Ave., Riverside, CA 92521, USA

**Keywords:** diabetes, technology, artificial intelligence, depression, diabetic foot ulcers, adherence

## Abstract

Diabetic foot ulcers, which are a common complication of diabetes, can have a negative impact on a person’s physical and mental health, including an increased risk of depression. Patients suffering from depression are less likely to keep up with diabetic foot care, thus increasing the risk of developing ulcers. However, with the use of artificial intelligence (AI), at-home patient care has become easier, which increases adherence. To better understand how new technologies, including machine learning algorithms and wearable sensors, might improve patient adherence and outcomes, we conducted a literature review of several sensor technologies, including SmartMat© and Siren Care© socks for temperature, SurroSense Rx/Orpyx© for pressure, and Orthotimer© for adherence. An initial search identified 143 peer-reviewed manuscripts, from which we selected a total of 10 manuscripts for further analysis. We examined the potential benefits of personalized content and clinician support for those receiving mobile health interventions. These findings may help to demonstrate the current and future utility of advanced technologies in improving patient adherence and outcomes, particularly in the context of diabetes management and the link between behavior and complications in diabetes, such as diabetic foot ulcers.

## 1. Introduction

Diabetic foot ulcers (DFU) can have a significant impact on a person’s physical and mental health, and numerous investigators have shown that there is a link between diabetic foot ulcers and depression [1,2]. People with diabetic foot ulcers may experience feelings of frustration, hopelessness, and low self-esteem due to their limited mobility and the challenges of managing their condition [3]. These negative emotions can contribute to a cycle of poor adherence to treatment and self-care, which can lead to the worsening of a foot ulcer and an increased risk in complications such as amputation [3].

Newer monitoring and wearable technologies, such as smart socks and insoles, can help improve patient adherence and outcomes by providing real-time data on foot pressure and temperature, alerting patients and healthcare providers to potential problems before they become serious, as highlighted in this manuscript. These technologies can also help people with diabetic foot ulcers to better understand and manage their condition, which can improve their overall quality of life and reduce the risk of depression. By providing ongoing support and monitoring, wearable technologies can help people with diabetic foot ulcers to feel more in control of their health and more confident in their ability to manage their condition.

Research focused on investigating the influence of comorbid depression on the development and advancement of foot ulcers has shown that depression can lead to healing delays and a triple rise in mortality risk within 18 months after the onset of a foot ulcer [3]. Depression has been linked to the reduced likelihood of healing and increased mortality in this high-risk population [4]. Self-inflicted behavioral habits that precipitate a diabetic foot complication often make real-time/long-term monitoring adherence a challenge. Patients with depression demonstrate a threefold decrease in adherence to treatment plans for chronic illnesses when compared to those without depression [5]. Professionals trained in the diagnosis and treatment of depression and diabetic foot ulcers should screen and evaluate patients. To be effective, patient home caregivers and significant others should be integrated into their respective treatment plans. With the global prevalence of diabetes increasing, it is imperative to identify previously undetected depression (mild, moderate, or severe) in individuals with diabetes [4]. This identification should occur either before complications arise or when complications are already present, as it is essential for enabling timely intervention [4]. Technology-based self-help treatment approaches for depression, such as the Internet and mobile platforms, enhance accessibility for individuals. These tools can be utilized conveniently in the comfort of one’s home, at a self-determined schedule and pace, without compromising privacy. Moreover, these solutions can provide valuable guidance and support to help patients improve their adherence to treatment. There are emerging technologies to help combat and/or help prevent the formation of diabetic foot ulcerations, which include monitoring systems that measure pedal temperature, mechanical plantar foot stress, and orthoses time adherence. Increasing adherence to diabetic foot care across different disease spectra involves essential components such as educating patients about their condition, emphasizing the significance of following prescribed treatments, and providing psychosocial support. These elements are considered fundamental pillars in promoting diabetic foot care adherence. The purpose of this review is to describe micro-climate regulating, stress monitoring on plantar tissue, and wear time of boots monitoring technologies that might help bridge the gap between patient adherence, patient and patient family actualization, and improved outcomes in those with diabetic lower extremity complications.

## 2. Diabetic Peripheral Neuropathy

Symptoms of diabetic peripheral neuropathy (DPN) include pain, tingling, and numbness [6]. The incidence of DPN ranges from 30 to 50% [7]. It often coexists with various other health conditions, such as cognitive impairment, depression, autonomic neuropathy, peripheral artery disease (PAD), nephropathy, retinopathy, cardiovascular disease, and medial arterial calcification [7,8]. Diagnosis of DPN is typically established through suggestive clinical symptoms and neurologic tests [9]. However, some patients with DPN may not exhibit overt signs of nerve damage, despite demonstrating evidence of neurologic deficits in nerve conduction studies (NCSs) or electromyography [9]. A significant characteristic of diabetic peripheral neuropathy involves diminished sensitivity to cold (threshold), heat, and touch [6]. There is a multitude of various causative factors that could be responsible for tissue damage and changes in sensation, including free radical damage, a metabolic by-product of cell glycolysis, edema, and a chronic inflammatory process [6]. The meticulous regulation of reactive oxygen species (ROS), including Nitric Oxide, Hydrogen Peroxide, etc., is essential for cellular homeostasis due to their critical involvement in normal cellular functioning. Additionally, these species are highly sensitive to glycemic control, further emphasizing their significance [10]. If glycemic control is not managed, it causes an imbalance and overproduction of harmful levels of reactive oxygen species (ROS) [10]. Excess ROS can lead to cellular proteins and membrane lipid damage, resulting in the accumulation of toxic peroxidation products that bind to the cellular nuclear material [10]. This process can trigger apoptosis, cause DNA damage, reduce axonal transport, and decrease the levels of neurotrophic factors responsible for maintaining normal nerve function [11].

Various macro-level factors that may heighten the risk of developing diabetic neuropathy are as follows [12]:Coronary artery disease;Increased triglyceride levels;Obesity;Smoking;High blood pressure.

Clinical recommendations advocate for enhancing the quality of diabetic foot care through the use of evidence-based risk assessments. Substantial evidence supports the early identification of ulceration risk through these evaluations of the diabetic foot, leading to a reduction in ulcer development [12]. It has been shown that patients who follow professional foot care recommendations (monitoring their foot temperatures, monitoring their foot pressures, or continuously wearing therapeutic footwear) have significantly better outcomes than those who do not follow or are unable to easily follow diabetic foot care protocols [13,14]. Patients suffering from depression are less likely to follow through with ulcer care; however, artificial intelligence (AI) monitoring has made it relatively easy for patients to keep up with adherence to care.

## 3. Methods

### 3.1. Search Strategy

Patients suffering from depression are less likely to keep up with diabetic foot care, thus increasing the risk of developing ulcers. However, with the use of artificial intelligence, at-home patient care has become easier, which increases adherence. We conducted a literature search on the database PubMed in order to find studies that relate to diabetic foot ulcers (DFUs) and artificial intelligence (AI). We set the time from 2005 to 2022. In order to narrow down the search for study, we looked into some of the leading causes of DFUs. We found them to be: plantar temperature, plantar pressure, and adherence to footwear. From there, we researched keywords that relate to AI, DFUs, and those causes. Here are the final search queries we did on PubMed: (Diabetic Foot ulcers) (Causes), (Diabetic Foot) (Wear Time) (Sensors), (Orthopedic footwear) (Temperature Sensor), (Diabetic Foot ulcers) (Plantar Temperature), (Diabetic Foot Ulcers) (SurroSense Rx©/Orpyx (Calgary, AB, Canada)), (Diabetic Foot Ulcer) AND (Sensor Socks), (Diabetic Foot Ulcers) AND (Remote Temperature Monitoring System; Podimetrics, Inc., Somerville, MA, USA). These findings can potentially demonstrate the utility of using robust technologies to improve patient adherence and demonstrate the potential benefits of integrating treatment pathways for the bidirectional relationship between depression and complications related to diabetes.

### 3.2. Analysis

The search yielded 134 peer-reviewed manuscripts. We filtered them down to works that mentioned plantar temperature, plantar pressure, and adherence. We also eliminated any papers that did not report quantitative data or include the use of a portable and wearable at-home monitoring device. After reading through the research, we narrowed down the type of artificial intelligence (AI) we wanted to focus on based on the risk factors that lead to ulcer formation (Figure 1). Table 1 summarizes the risk factors focused on in this paper. We analyzed SmartMat© and Siren Care© sensor socks for temperature, SurroSense Rx/Orpyx© for pressure, and Orthotimer© and SmartBoot for adherence.

## 4. Results

### 4.1. Diabetic Foot Micro-Climate Regulating Technology

The foot micro-climate refers to intrinsic and extrinsic factors that regulate pedal health, including temperature, pressure, humidity, shear, and stress. Studies conducted by Armstrong et al. revealed that patients who had previously experienced ulceration demonstrated elevated local skin temperatures during their follow-up visits before experiencing re-ulceration [15], thus highlighting the potential benefit of assessing skin temperature measurements to predict ulcer formation. Despite the growing evidence of skin temperature monitoring, there are no current diagnostic algorithms in any current clinical practice guidelines. One significant challenge is patient adherence to daily temperature measurements, including any potential learning curve in using these technologies. Some patients may find it difficult to learn how to use certain temperature monitoring technologies. This learning curve might deter a lot of patients from wanting to keep up with regular temperature monitoring. Additionally, some patients might also find incorporating regular temperature monitoring into their daily routine difficult. This issue can also increase in patients suffering from depression. Depression can have a significant impact on a patient’s ability to keep up with care plans put in place by their physicians, including adherence to daily temperature measurements. The lack of motivation, low energy levels, and a sense of hopelessness caused by depression can make it difficult for patients to want to continue to engage in monitoring their health [23]. Patients suffering from depression might also have difficulty with memory and concentration, which might also be a reason for not keeping up with their regular temperature readings, as they might forget to monitor their temperature. To address these challenges, healthcare providers have started to incorporate easy-to-use AI technology to make temperature monitoring easy and accessible for all types of patients.

Frykberg and their coworkers [16] designed SmartMat, an AI-centric device that has three key performance indicators: assessing the mat’s efficacy in identifying plantar DFU at an early stage, examining participant adherence to its usage over time, and gaining insights into participant perceptions regarding potential benefits and user-friendliness. SmartMat uses specialized sensors that integrate an image processing system to compare temperatures between normal and abnormal feet [16]. This mat was designed to be used at home and only requires the participant to step on the mat with both feet for 20 s [16]. The collected data are subsequently uploaded to a cloud storage system through a built-in cellular component integrated within the mat. The research showed that by setting the temperature to an asymmetry of 2.22 °C, the mat was able to accurately predict 97% of DFUs with only a false positive rate of 57% [16]. The mat also had a high adherence rate of 86% with the use of the mat averaging 3 days per week [16]. Patients throughout the study documented the ease of using the mat. Making the mat extremely accessible and easy to use helped increase the adherence to monitoring plantar temperature. Patients suffering from depression might appreciate the simplicity and continue incorporating it into their daily care treatment. The high success rate and adherence to SmartMat seem promising for effectively incorporating this in patient care to reduce the risk of re-ulceration [16].

There are other wearable technologies that help improve adherence to the daily monitoring of plantar temperature. Siren Care utilizes a specialized sock with embedded sensors that continuously monitor daily plantar foot temperatures [17]. This sock is embedded with six sensors—at the metatarsal points, midpoint, and heel—that read the patient’s plantar temperature in 10 s intervals [17]. These specialized sensors monitor daily activity levels while continuously aggregating plantar foot data points (temperatures). The microsensors in the socks are paired with a cellphone via Bluetooth and show any increase in plantar temperature on the mobile device [17]. The pilot study conducted showed that the temperature measured was within 0.2 °C of the clinically observed temperatures [17]. Based on the comfort and simplicity of the socks, adherence was noted to be higher [17]. This technology may assist in the improvement of ulceration in patients that are at high risk of DFU.

Incorporating sensors into socks can be a solution to the challenge of patient adherence, especially for those suffering from depression. By removing the need for patients to remember to take their own temperature and instead having a sensor attached to their socks, the cognitive load on patients is greatly reduced. This is especially important for patients with depression who may struggle with attention and memory [24]. Simplifying healthcare plans can help these patients manage and follow through with their treatment plans. With the sensor already in place, depressed patients are more likely to stick with their healthcare plans without the added stress of having to remember to take their own plantar temperature. Overall, incorporating sensors into socks can provide a convenient and effective method to improve patient adherence and health outcomes for those with depression. The Siren Care socks have the potential to mitigate the risk of ulcer development, thus reducing the likelihood of severe complications, like foot amputations.

In order for the treatment of diabetic foot ulcers to be effective, patients must monitor their plantar temperature daily. Different types of AI incorporated in sensors are making it easier for patients to stick to their care regimen, thus increasing adherence. A study conducted on PodoTemp, another temperature monitoring technology, compared both the platform’s accuracy and consistency in repeated testing, as well as the usability and adherence in patients’ home environments [25]. PodoTemp is equipped with 120 temperature sensors embedded in each foot [25]. Each sensor is used to individually measure the temperature of the foot while the patient stands on the platform for 40 s [25]. The platform was originally designed to be easy to use and incorporate into any at-home patient DFU care. The built-in algorithm on the platform helps patients gather their plantar temperature and analyze the measurements without needing any help [25]. This would be especially helpful to incorporate in patients who are at risk of developing ulcers and suffering from depression.

The first part of the study by Veneman et al. was conducted on patients who lost all protective sensation due to peripheral neuropathy. Each participant stood on the platform for 40 s to get a reading of the plantar temperature [25]. In order to accurately compare the results, TempTouch (Diabetica Solutions, San Antonio, TX, USA), an infrared handheld skin thermometer was used. The TempTouch was used as a reference because it is the most commonly used equipment in studies revolving around foot temperature monitoring [15,26,27,28]. The first part of the study showed the accuracy of the PodoTemp measurements, as the results were very similar to the handheld infrared skin thermometer. The second part of the study aimed to test the usability and the increase in adherence to the platform. Over the course of two weeks, participants were asked to use the PodoTemp platform in their home environment [25]. Participants were then asked to complete a survey about their experience with the platform. A total of 87% of the participants expressed that the platform was easy to use, and 67% of the participants were motivated to continue to monitor their plantar temperature on a daily basis [25]. It was reported that participants felt that the platform was quicker to use and required little to no action on their end, thus increasing their desire to continue monitoring their temperature on a daily basis. The platform was designed in order to be easy to use and mass-produced at a relatively low cost. The platform’s accurate ability to execute and analyze temperature measurements without being difficult on patients is a major advantage, especially for those suffering from depression, compared to many other available foot temperature monitoring devices—like the handheld thermometer—and can potentially be incorporated for use at patients’ homes to treat diabetic foot ulcers.

As demonstrated, SmartMat, Siren Care, and PodoTemp all play a vital role in the care and prevention of diabetic foot ulcers. Each technology brings a unique approach to monitoring plantar temperature, as demonstrated in Table 2. These distinct features contribute to enhanced patient care and a reduced risk of ulcer formation. While the functionalities and data analysis methods may vary among these technologies, their shared objective is to facilitate early detection of foot ulcers and enable timely intervention, ultimately improving patient care and mitigating the likelihood of additional complications.

### 4.2. Monitoring Stress on Plantar Tissue

The nefarious effects of diabetic peripheral neuropathy can sometimes leave repetitive foot stress vectors unaddressed in patients with diabetes [18]. Repetitive microtrauma along prominent osseous foot structures inherently causes tissue breakdown [18]. Ulcer formation can be attributed to sequelae related to sensory, motor, and autonomic neuropathy [18]. Patients with diabetes also tend to have poor tissue quality, making their feet more susceptible to tissue breakdown [18]. This susceptibility highlights a critical component in the management of plantar foot stress vectors in the neuropathic diabetic foot. One of the main tenets of diabetic foot care is the utilization of proper offloading devices [18]. Existing offloading devices lack a feedback mechanism to address tissue breakdown. SurroSense Rx© (Orpyx Medical Technologies, Calgary, AB, Canada) is an AI device that has the ability to monitor plantar foot stress. The core of this technology is within a customized insole (with integrated sensors) that connects wirelessly to a smartwatch. The SurroSense Rx© can provide real-time alerts to patients about plantar pressure distribution [19]. Table 3 provides detailed parameters of SurroSense Rx©.

A recent study demonstrated that participants in remission wore the New Balance 929 Diabetic Walking shoe with two pressure-sensing insoles and a smartwatch [19]. Each insole was embedded with eight pressure sensors—three on the metatarsal heads, two on the lateral plantar surface, one on the heel, one on the great toe, and one in the distribution of the lateral toes [19]—as shown in Figure 2. The smartwatch sent alerts to the participant if greater than 95% of the measured pressure was above the determined threshold of 35–50 mmHg for more than 15 minutes [19]. The watch also measured the success rate of the patient’s response. A successful response meant the pressure was offloaded in less than 20 min after the alert was sent. Patients on average wore the insoles for about 5.38 ± 3.43 h per day and got about 3.38 ± 3.81 alerts per day throughout the study [19]. The study revealed that an alert is required every two hours for patients to respond to any potential issues [19]. The smart insole’s alert feedback is effective for high-risk diabetic patients and could help reduce the pressure on high-risk areas of the foot, which will help reduce repeated stress on the plantar tissue.

The smart insoles’ alert feedback system provides a major benefit to patients with depression. Individuals with depression often struggle with memory and cognitive functioning, and the repeated reminders sent by the insoles can be especially helpful. By sending regular alerts to offload pressure on the plantar region, patients no longer have to constantly remember to take care of their health and offload pressure. This reduction in stress and cognitive load can significantly improve patients’ abilities to adhere to their treatment plans, ultimately decreasing the risk of ulcer development.

A recent, randomized proof-of-concept from two multidisciplinary outpatient diabetic foot clinics in the UK was randomly assigned to either an intervention (SurroSense Rx© insole system) or control. The intervention group received audiovisual and vibrational alerts from the smartwatch, encouraging the patient to offload by walking or removing the weight from the affected foot [20]. Once the pressure was offloaded, the alert on the device cleared, and patients were able to resume normal activity. The control group received no alerts, regardless of plantar pressure being detected. The pressure detected was considered high if 95–100% of the readings were above 35 mmHg and low if 0–34% of the readings were above 35 mmHg [20]. Based on the findings from the study, ulcer incidence was reduced by 86% in the intervention group versus the control group [20]. The results from this study infer that continuous plantar pressure monitoring and dynamic offloading guidance can potentially lead to a reduction in diabetic foot ulcer site recurrence.

### 4.3. Monitoring Wear Time of Boots

Another key factor leading to diabetic foot ulcer recurrence is patient adherence to prescribed diabetic foot care [21]. Adherence is oftentimes defined as a patient’s behavior that corresponds directly to the agreed-upon recommendations from their respective healthcare provider [21]. There is a direct correlation between poor adherence to diabetic foot care and ulceration recurrence [21]. However, new technologies can now help with monitoring footwear adherence, thus potentially improving patient outcomes. Orthotimer© (Balingen, Germany) is a temperature-sensing modality used to monitor how long patients are using their prescribed footwear [21].

Lutjoboer et al. conducted a recent study using a temperature monitoring system for patient adherence. Ten healthy participants were monitored over a duration of 48 h using the sensor in their footwear. The technology utilizes a microsensor to record temperature every 15 min [21]. If the leg was visible in the photograph, the patient was noted to be adherent (wearing the device). The average footwear measured using the camera was 8.10 h per day, and the average footwear measured using the sensor was 8.16, 8.86, and 4.91 per day [21]. The similarities in the results proved how effective the sensor was in determining whether the footwear was in use or not. The accuracy of the results demonstrates how using the Orthotimer© could be an effective way of determining how long a patient was wearing the required footwear. This can help doctors adjust their course of patient care and help increase patient adherence to the footwear [21].

Another effective way to help increase patient adherence when it comes to the required footwear is SmartBoot [30]. This boot uses a smart offloading system in order to remotely monitor a patient’s real-time adherence to the required footwear. The smart offloading boot is used with a smartwatch and stores the data on a cloud dashboard in order to collect a patient’s adherence and activity [30]. In order to improve the effectiveness of healing through boots and preventing DFUs, adherence to offloading devices needs to be monitored. Currently, there are four different ways to promote offloading [31]. These methods include wearing different footwear, like shoes and insoles; surgery, like silicone injections; offloading devices, like removable and non-removable devices; and offloading techniques, like wheelchairs and bed rest [31]. Although these techniques might be beneficial in protecting patients from developing diabetic foot ulcers, they are not as effective in increasing adherence [31]. One way that clinicians found around this issue was by introducing a non-removable offloading device [30]. Even though this offloading device is very good at treating foot ulcers, it comes with severe limitations [30]. Having a non-removable offloading device hinders a lot of patients’ daily activities. Although this helped increase adherence to footwear devices, patients were not very satisfied and comfortable with the treatment. Having a smart offloading device was one way to address this issue. This device uses the smartwatch to send alerts to increase adherence, thus reminding the patients to continue wearing their boots [30]. The SmartBoot is able to remotely monitor a patient’s weight bearing activity. A study conducted by Park et al. reported that patients were extremely satisfied and comfortable with the Smartboot [30]. They felt that they were able to do their daily activities and still continue wearing the device, thus increasing adherence.

Orthotimer© and SmartBoot are invaluable technologies that play a crucial role in promoting adherence and aiding in the prevention of diabetic foot ulcers. Table 4 illustrates their distinct approaches to monitoring and enhancing adherence. Despite their differing functionalities and data analysis methods, both technologies serve as essential tools for mitigating the risk of ulcer formation.

## 5. Summary/Conclusions

Thinking creatively about targeting both diabetic foot complications and behavior change as it relates to treatment adherence is vital for overall patient outcomes. In order for these digital health technologies to optimize their effectiveness, they must address diabetic-related conditions in a more holistic way that promotes patient engagement while also monitoring adherence. Encouraging patient engagement is particularly important in a diabetic patient who suffers from clinical depression. Technologies that are able to address the intersecting connection between pathology and psychosocial variables in patient care could potentially set the stage for future treatment algorithms. Adhering to clinical practice guidelines, routine assessment, screening, and treatment of depression in patients with diabetes is recommended. Implementing advanced technologies that integrate artificial intelligence can prevent diabetic-related complications, thus improving quality of life and potentially reaching a decrease in both patient and health service costs by engaging in a less expensive and more accessible treatment.

This literature review highlights the significant impact of diabetic foot ulcers on physical and mental health, particularly in relation to the link between DFUs and depression. The emotional toll of limited mobility and the challenges of managing the condition can lead to frustration, hopelessness, and low self-esteem among individuals with DFUs. These negative emotions contribute to a poor adherence to treatment and self-care, resulting in worsening foot ulcers and increased preventative complications, such as amputations.

This review emphasizes the potential of smart wearable technologies, including smart socks and insoles, in reducing the risk of ulcer formation and improving patient adherence. These technologies provide real-time data on plantar pressure and temperature, enabling early detection of potential ulcer formation and facilitating timely intervention. Moreover, they empower patients with DFUs to better understand and manage their condition, leading to an improved quality of life and reduced risk of depression.

Comorbid depression has been found to significantly impact the incidence and progression of foot ulcers, leading to delayed healing and increased mortality rates. Individuals with depression are less likely to adhere to medical regimens and require integrated treatment approaches that involve professionals trained in both depression and diabetic foot ulcers. Engaging patient home caregivers and significant others in the treatment plan is also crucial for effective management.

This review further highlights the value of technology-delivered self-help treatment approaches for depression, which can be easily accessed and utilized in the privacy of a patient’s home. These approaches overcome transportation-related obstacles and provide cost-effective solutions. The incorporation of AI technologies, such as continuous plantar temperature monitoring systems and sensor-equipped socks, addresses the challenges of patient adherence, particularly for individuals with depression. These technologies simplify monitoring routines, reduce cognitive load, and improve patient engagement with their healthcare plans.

Overall, the literature supports the use of robust technologies to enhance patient adherence and improve outcomes in individuals with diabetic foot complications. The integration of AI monitoring holds promises for reducing the incidence of DFUs, promoting timely intervention, and improving overall diabetic foot care adherence. By embracing these advancements, healthcare providers can optimize patient care, prevent complications, and ultimately improve the lives of individuals who are at risk of or currently suffering from diabetic foot ulcers. Further studies into micro-climate regulation, stress monitoring on plantar tissue, and the monitoring of the wear time of boots with integrated sensors will highlight the significance of integrating AI tools into patient care regimens.

## Figures and Tables

**Figure 1 sensors-23-06898-f001:**
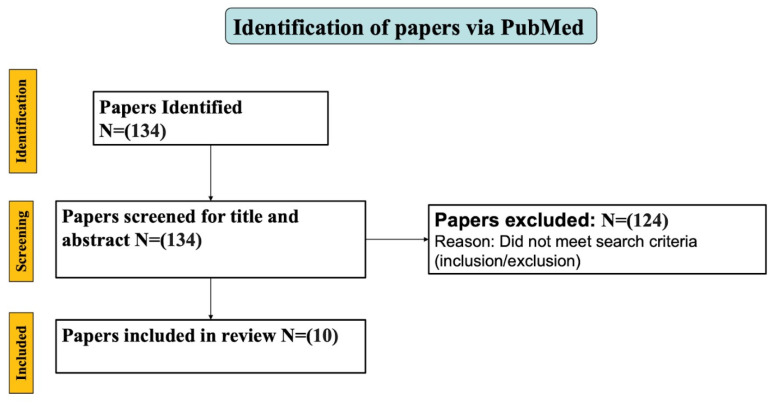
Flowchart of how the papers were filtered.

**Figure 2 sensors-23-06898-f002:**
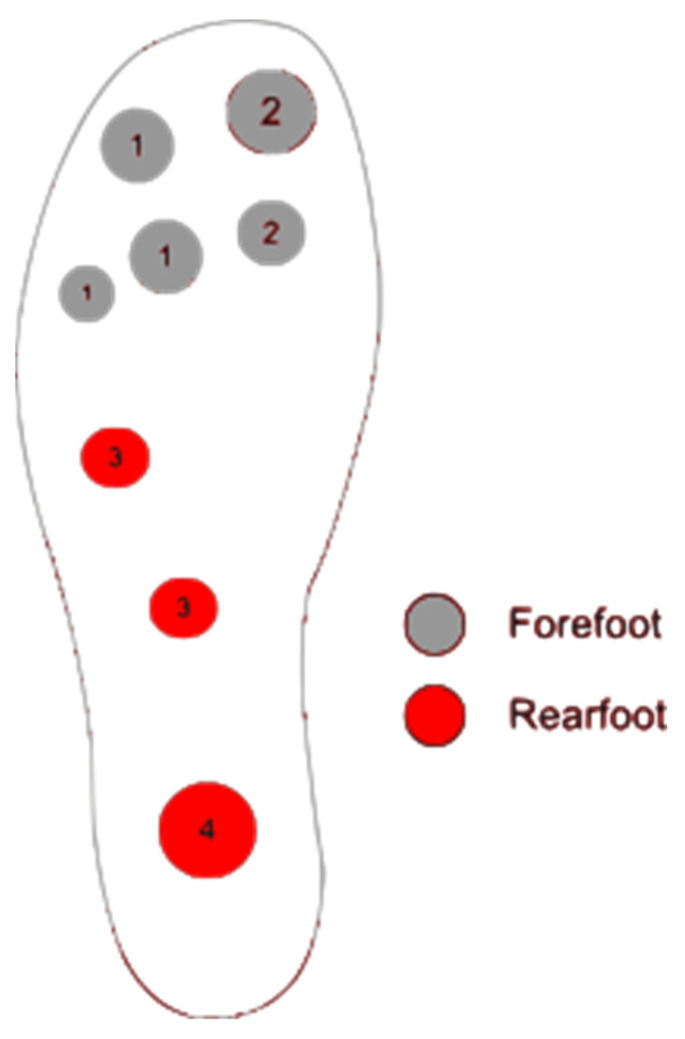
Location of the eight pressure sensors on the insole. Numbers indicating the different sensors and different colors are for the forefoot and rearfoot [19,29].

**Table 1 sensors-23-06898-t001:** Risk factors for DFUs with current management and AI management.

Risk Factors	Current Management	Issues with Current Management	Sensors/Devices	Potential Impact on Prevention	References
Lesions occurring prior to ulceration, arising from irregularities in temperature	Hand-held thermometer device	Might be difficult and time consuming for patients	Continuous at-home plantar temperature monitoring system	Minimizing the risk of potential sites susceptible to ulcer development	[15,16,17]
Elevated plantar pressure	Offloading footwear	Irregular adherence	Continuous at-home plantar pressure monitoring system with patient feedback on a mobile device	Improvement of timely offloading and potentially reducing ulcer occurrence	[18,19,20]
Irregular adherence	Total Contact Cast (TCC)	Prevent daily wound inspection and dressing changes	Monitoring adherence with temperature sensors and patient feedback on a mobile device	Prolongs patient adherence with diabetic orthopedic wear, which can potentially reduce ulcer recurrence	[21,22]

**Table 2 sensors-23-06898-t002:** Comparison of functionality, data collection and analysis, user interface, and intervention/alerting for SmartMat, Siren Care, PodoTemp, and TempTouch.

Technology	Functionality	Data Collection and Analysis	User Interface	Intervention and Alerting	References
SmartMat	Sensors embedded in a mat that integrates an image processing system to compare temperatures between normal and abnormal feet	After 20 s, the data are collected and then uploaded onto a cloud using a cellular component that is already in the mat	Temperature is not displayed on the mat; results are uploaded to a server	No alerts given; however, the temperature measurements will help physicians make informed decisions regarding intervention	[16]
Siren Care	Temperature sensors embedded in socks to detect change in temperature throughout the day	Collects the temperature data and sends them to a smartphone for monitoring and analysis	Uses a smartphone to show temperature data and alerts	Smartphone application provides alerts if there is a temperature change and will allow for timely intervention	[17]
PodoTemp	Total of 120 temperature sensors embedded for each foot on a platform that measures temperature differences between each foot	Provides instant readings for analysis after 40 s	Displays the temperature on the device	No alerts given; however, the temperature measurements will help physicians make informed decisions regarding intervention	[25]

**Table 3 sensors-23-06898-t003:** Comparison of functionality, data collection and analysis, user interface, and intervention/alerting for SurroSense Rx©.

Technology	Functionality	Data Collection and Analysis	User Interface	Intervention and Alerting	References
SurroSense Rx©	Insoles embedded with eight pressure sensors per foot; measures pressure on plantar side of feet	Collects pressure readings and sends it to smartwatch if an alert is needed	Displays alerts and readings on smartwatch	Wirelessly connects to a smartwatch to send real-time alerts to patients about plantar pressure distribution	[19]

**Table 4 sensors-23-06898-t004:** Comparison of functionality, data collection and analysis, user interface, and intervention/alerting for Orthotimer© and SmartBoot.

Technology	Functionality	Data Collection and Analysis	User Interface	Intervention and Alerting	References
Orthotimer©	Microsensor embedded in footwear to monitor how long patients are using the prescribed footwear	Collects temperature every 15 min	Patients cannot see collected temperature	No alerts given; however, the temperature measurements will help physicians make informed decisions regarding intervention and	[21]
		computer		whether patients are adhering to the prescribed footwear	
SmartBoot	Uses a smart offloading system in order to monitor real-time adherence to prescribed footwear and monitor patients’ weight bearing activities	Paired with a smartwatch to collect data on patient’s adherence and stores them on a cloud dashboard	Alerts shown on the smartwatch	The smartwatch sends alerts to remind patients to continuously wear the prescribed footwear	[30]

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
