# Peer review of "Adherence and the Diabetic Foot: High Tech Meets High Touch?"

_sensors, 2023, doi:10.3390/s23156898_

Round 1

Reviewer 1 Report

1.     The authors mentioned they have already reviewed 143 peer-reviewed manuscripts; however, the total number of references for this work was only 29, which is a small number for a review article.

2.     The authors should pay more attention to the order of references in the main text. For example, there was not any reference before reference #5, which suddenly appeared on line #56 (page 2). Similarly, there were references #7-9 before the occurrence of reference #10 (line 90, page 2). Strangely, references #18-22 appear on page 3 before reference #16 (page 4).

3.     The style of this article looks like a research paper, including some typical sections: introduction, methods, results, and conclusions. In addition, a "Review" should provide quality references and new insights for the reader.

4.     The content of this article is rather poor when it has only 1 table and 1 figure in order to summarize previous work.

5.     In general, the article requires extensive grammatical and formatting corrections. Authors must carry out a complete revision of the article writing. Many sentences in the text are too long and therefore confusing to the reader. Please try to shorten those to make digesting the content easier.

Author Response

Our response to the reviewers comments:

  1. The authors mentioned they have already reviewed 143 peer-reviewed manuscripts; however, the total number of references for this work was only 29, which is a small number for a review article.
  • Yes, 134* papers were identified and reviewed however 124 were excluded leaving only 10 that were actually included, the appropriate number of references for a systematic review depends on various factors, including the scope of the review, the availability of relevant studies, and the quality of the evidence. Considering the novelty of artificial intelligence as it pertains to the diabetic foot, 29 references is sufficient.
  • Here is an example of a recent systematic review that only included 36 references. Again, just to highlight the subjectivity of the "appropriate" number needed for systematic reviews. 

https://journals.lww.com/md-journal/Fulltext/2023/03100/Lisfranc_and_Chopart_amputation__A_systematic.65.aspx

  1. The authors should pay more attention to the order of references in the main text. For example, there was not any reference before reference #5, which suddenly appeared on line #56 (page 2). Similarly, there were references #7-9 before the occurrence of reference #10 (line 90, page 2). Strangely, references #18-22 appear on page 3 before reference #16 (page 4)
  • Noted will be revised accordingly. 
  • All references have been moved around to follow numerical order throughout the paper
  1. The style of this article looks like a research paper, including some typical sections: introduction, methods, results, and conclusions. In addition, a "Review" should provide quality references and new insights for the reader.
  • As noted above, The appropriate number of references for a systematic review depends on various factors, including the scope of the review, the availability of relevant studies, and the quality of the evidence. Considering the novelty of artificial intelligence as it pertains to the diabetic foot, 29 references is sufficient.
  1. The content of this article is rather poor when it has only 1 table and 1 figure in order to summarize previous work.
  • Noted will integrate more figures into the manuscript.
  • Added another figure indicating where each sensor is located for SurroSense Rx©. This makes a total of two figures and one table.
  1. In general, the article requires extensive grammatical and formatting corrections. Authors must carry out a complete revision of the article writing. Many sentences in the text are too long and therefore confusing to the reader. Please try to shorten those to make digesting the content easier.
  • Thank you for taking the time to review our manuscript. We acknowledge the challenges faced in trying to convey complex ideas in a clear and concise manner, which resulted in many long and confusing sentences. However, we are committed to improving the quality of our writing and will make every effort to revise the article accordingly. 
  • Due to the vague nature of this recommendation, we kindly request an extension if we are to completely revise the syntax of this manuscript in its entirety. Although it should be noted this comment is in stark contrast to Reviewer 2's perspective stating he/she felt "This is a well written review paper and was an interesting read."
  •  Please feel free to highlight and send back which sentences need to be modified or shortened.
  • We went through the paper and tried shortening sentences that seemed too long. We also added a couple paragraphs on how the devices would be easier for people who suffer from depression due to certain symptoms they experience.

Reviewer 2 Report

This is a well written review paper and was an interesting read. There were a few minor points related to referencing (I think the pdf version may have got rid of some earlier refs) and some minor comments, but overall a very good paper.

Specific comments/suggested corrections are:

* line 27 - could you provide a few references to back up this statement

* Line 25-31 - please can you provide references to back up these statements

* line 33 and 34 - please provide refs these technologies

* line 43-46 which studies - please provide references

* line 50 - please provide a ref

* line 63-65 please provide refs of technologies

* line 72 please provide ref for DPN prevalence

* line 75 - there appears to be a small 6 - is this a ref? Has there been an issue with refs in pdf version of manuscript? Is this why refs are missing from earlier lines?

*Line 101-104 please provide ref for statement

* Line 104-106 - can you expand this statement (add another sentance to explain a little further) and provide ref if possible. I think IA and adherence is one of  the key messages of your paper

* line 109-11 might be better in the previous section. Also do you mean AI has the potential? You have not yet told the reader the evidence for this (which I think is the premise of your review). Suggest you might rephrase to "potential to increase adherence...."

* Table 1 - good table and content. Is it possible not to have words split by hyphens on multiple lines - could you reformat to remove hyphens and not split words across lines

* Line 136 - no figure number for diagram. Is this a registered PRISMA review? If so please include these details in the text.

* line 141 - can you change to shear stress and normal stress or shear pressure and normal pressure to be more specific

* line 149 - please change to "a Smart Mat, a temperature measurement AI-centric device...."

* line 249 - suggest you use another word for boots - can you be more specific or do you mean offloading footwear in general?

* line 290 - referring to line 141 you don't find anything on shear stress or humidity. Could you add a line at the end here or at the beginning of the section 4 to state that commercial devices currently don't exist to measure these parameters in-shoe

Overall good paper making a strong contribution - well done to the authors. I hope these comments suggestions will improve their paper

Author Response

* line 27 - corrected

* Line 25-31 - corrected

* line 33 and 34 - corrected

* line 43-46 : corrected and reference provided

* line 50 - corrected

* line 63-65 corrected

* line 72 corrected

* line 75 - there appears to be a small 6 - is this a ref? Has there been an issue with refs in pdf version of manuscript? Is this why refs are missing from earlier lines? Answer: very possible; not sure what happened.

*Line 101-104 corrected

* Line 104-106 - can you expand this statement (add another sentance to explain a little further) and provide ref if possible. I think IA and adherence is one of  the key messages of your paper: See modification in attached docuement

* line 109-11 might be better in the previous section. Also do you mean AI has the potential? You have not yet told the reader the evidence for this (which I think is the premise of your review). Suggest you might rephrase to "potential to increase adherence...." corrected

* Table 1 - good table and content. Is it possible not to have words split by hyphens on multiple lines - could you reformat to remove hyphens and not split words across lines: I was unable to remove hyphens as it was automatically placed due to size of font (final editor may be able to correct?)

* Line 136 - no figure number for diagram. This is not a registered PRISMA; figure line noted

* line 141 - can you change to shear stress and normal stress or shear pressure and normal pressure to be more specific: corrected

* line 149 - please change to "a Smart Mat, a temperature measurement AI-centric device...." corrected

* line 249 - suggest you use another word for boots - can you be more specific or do you mean offloading footwear in general? corrected

* line 290 - referring to line 141 you don't find anything on shear stress or humidity. Could you add a line at the end here or at the beginning of the section 4 to state that commercial devices currently don't exist to measure these parameters in-shoe corrected in section 4 see attached document 

Round 2

Reviewer 1 Report

The authors have tried to respond in detail to the raised issues and made considerable efforts to revise the manuscript. The manuscript was modified in several places, which improved the manuscript. However, the reviewer is not convinced that the revised manuscript is adequate for publication in its present form. The scientific quality of this paper should be improved; thus, I recommend major revisions before the final decision.

Round 3

Reviewer 1 Report

The authors have shown a great effort to address all comments raised by the reviewer. The quality of this manuscript has improved now and could be considered to meet the standard of Sensors journal. 

However, by using Turnitin, the reviewer detects a surprising rate of plagiarism (31%). The authors should check and revise the whole revision to reduce the plagiarism rate. Hence, a major revision is absolutely required before publication. The reviewer is enclosing the report of plagiarism below. Please check and revise the manuscript.

Some other comments:

#151: "Armstrong et al. (2007)" should be numbered instead of listed like this.

#112: The authors should highlight that the short-form "AI" is shortened from Artificial Intelligence for its first appearance.

#287: please check the accuracy of this value "5.38±3.43 hours per day" and "3.38±3.81 alerts per day

#386: the authors mentioned that: "The review emphasizes the potential of AI monitoring and wearable technologies"; however, the review only focuses on smart wearable technologies. Please make a clear discussion on AI monitoring in the revision.

Author Response

The authors have shown a great effort to address all comments raised by the reviewer. The quality of this manuscript has improved now and could be considered to meet the standard of Sensors journal. However, by using Turnitin, the reviewer detects a surprising rate of plagiarism (31%). The authors should check and revise the whole revision to reduce the plagiarism rate. Hence, a major revision is absolutely required before publication. The reviewer is enclosing the report of plagiarism below. Please check and revise the manuscript.

We went through and revised anything that was flagged for plagiarism.

Some other comments: #151: "Armstrong et al. (2007)" should be numbered instead of listed like this.

We changed it to numbering

#112: The authors should highlight that the short-form "AI" is shortened from Artificial Intelligence for its first appearance.

We highlighted in abstract and introduction.

#287: please check the accuracy of this value "5.38±3.43 hours per day" and "3.38±3.81 alerts per day"

The source listed these values as part of their results.

#386: the authors mentioned that: "The review emphasizes the potential of AI monitoring and wearable technologies"; however, the review only focuses on smart wearable technologies. Please make a clear discussion on AI monitoring in the revision.

We fixed this issue.